# LLMScore: Unveiling the Power of Large Language Models in Text-to-Image Synthesis Evaluation

**Yujie Lu**[1]*, **Xianjun Yang**[1], **Xiujun Li**[2], **Xin Eric Wang**[3], **William Yang Wang**[1]

[1]University of California, Santa Barbara
[2]University of Washington      [3]University of California, Santa Cruz
https://github.com/YujieLu10/LLMScore

## Abstract

Existing automatic evaluation on text-to-image synthesis can only provide an image-text matching score, without considering the object-level compositionality, which results in poor correlation with human judgments. In this work, we propose LLMScore, a new framework that offers evaluation scores with multi-granularity compositionality. LLMScore leverages the large language models (LLMs) to evaluate text-to-image models. Initially, it transforms the image into image-level and object-level visual descriptions. Then an evaluation instruction is fed into the LLMs to measure the alignment between the synthesized image and the text, ultimately generating a score accompanied by a rationale. Our substantial analysis reveals the highest correlation of LLMScore with human judgments on a wide range of datasets (Attribute Binding Contrast, Concept Conjunction, MSCOCO, DrawBench, PaintSkills). Notably, our LLMScore achieves Kendall's $\tau$ correlation with human evaluations that is $58.8\%$ and $31.2\%$ higher than the commonly-used text-image matching metrics CLIP and BLIP, respectively.

## 1  Introduction

In recent years, research in text-to-image synthesis has made significant progress [9, 11, 38, 43]. However, evaluation metrics have lagged behind due to challenges such as accurately capturing composite text-image alignment (e.g. color, counting, location) [47], interpretably producing the score, and adaptively evaluating with various objectives.

Established evaluation metrics for text-to-image synthesis like CLIPScore [17] and BLIP [24], while widely used and highly effective [20, 36], have encountered challenges when it comes to capturing object-level alignment between text and image [12, 26]. Figure 1 illustrates an example from the Concept Conjunction dataset [12], given the text prompt ''A red book and a yellow vase'', the left image aligns with the text prompt, while the right image *fails to generate a red book, and the correct color for the vase, also with an extra yellow flower*. Human judges can make the correct and clear assessment (1.00 v.s. 0.45/0.55) of these two images on both overall and error counting objectives, while the existing metrics (CLIP, NegCLIP [51], BLIP) predicts similar scores for both images, failing to distinguish the correct image (on the left) from the wrong one (on the right). Furthermore, these metrics provide a single, non-interpretable score, obscuring the underlying reasoning behind the alignment of the synthesized images with the given text prompts. Additionally, these model-based metrics are static, unable to follow varied guidelines that prioritize different objectives of the text-to-image evaluation. For example, the evaluation can range from accessing image-level semantics (Overall) to finer object-level details (Error Counting). These issues hinder the existing metrics from aligning with human evaluations.

---

*Corresponding Author: yujielu10@gmail.com

37th Conference on Neural Information Processing Systems (NeurIPS 2023).

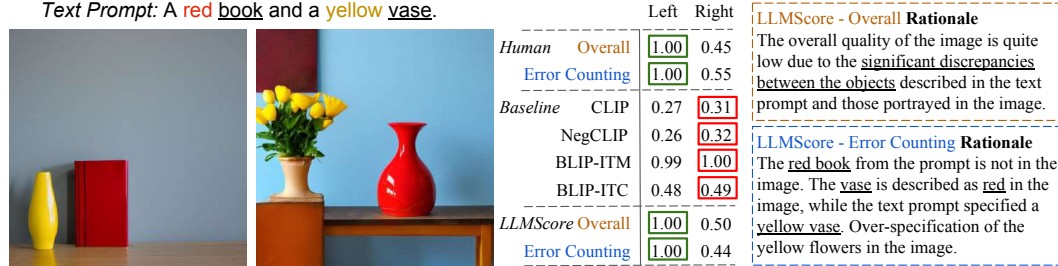

Figure 1: The two images are generated using Stable-Diffusion-2 based on the text prompt sampled from the Concept Conjunction dataset. *Baseline* section shows the scores from the existing model-based evaluation metrics, *Human* section is the rating score from the human evaluation, *LLMScore* section is our proposed metric. The right column also shows the rationale generated by LLMScore.

In this paper, we introduce LLMScore, a novel framework to evaluate text-image alignment in text-to-image synthesis by unveiling the powerful reasoning abilities of large language models (LLMs). Our inspiration stems from the human approach of measuring text-image alignment, which involves checking the correctness of objects and attributes specified in the text prompt. With the incredible text reasoning and generation ability of LLMs, LLMScore can imitate the human evaluation process to access compositionality at multi-granularity and produce alignment scores with rationales, delivering more insight into the model performance and reasons behind the scores.

To enhance the evaluation of composite text-to-image synthesis, our LLMScore elicits grounding visio-linguistic information from vision and language models and LLMs, thereby capturing multi-granularity compositionality in the text and image. Specifically, our approach leverages vision and language models to transform the image into multi-granularity (image-level and object-level) visual descriptions, which allows us to capture the compositional aspects of multiple objects in the language format. Then we concatenate these descriptions with text prompts and feed them into large language models (LLMs, for example, GPT-4 [32]) to reason the alignment between text prompts and images. While existing metrics struggle to capture compositionality, our LLMScore captures the object-level alignment between text and image, producing scores that are significantly correlated with human evaluation, complete with reasonable rationales (Figure 1). In addition, our LLMScore can adaptively follow various guidelines (overall or error counting) by simply customizing the evaluation instruction for LLMs. For example, we can access the overall objective by prompting the LLMs with the instruction ''Rate the overall alignment of text prompt and image.'' or validate error counting objective with the instruction ''How many compositional errors are in the image?''. And we explicitly include guidance on error types of text-to-image models in the evaluation instruction to keep the LLM's decision deterministic. This flexibility empowers our framework as a versatile tool for a wide range of text-to-image tasks and various evaluation guidelines.

We validate the effectiveness of LLMScore through extensive experiments, demonstrating its alignment with human judgments without any demand for additional training. Our experimental setup contains state-of-the-art text-to-image models such as Stable Diffusion [38] and DALL·E [37], evaluated over diverse datasets, including prompt datasets for general purpose (MSCOCO [28], Draw-Bench [42], PaintSkills [8]) and for compositional purposes (Abstract Concept Conjunction [12], Attribute Binding Contrast [12]). Our LLMScore achieve the highest human correlation across all datasets. On compositional datasets, we achieve $58.8\%$ and $31.27\%$ higher Kendall's $\tau$ over widely used metrics CLIP and BLIP respectively.

To sum up, we present LLMScore, the first attempt to unveil the power of the large language models for text-to-image evaluation, in particular, our paper makes the following contributions:

- We propose LLMScore, a new framework to evaluate the alignment between text prompts and synthesized images in text-to-image synthesis, offering scores that accurately capture multi-granularity compositionality (image-level and object-level).

- Our LLMScore produces accurate alignment scores with rationales following various evaluation instructions (overall and error counting).

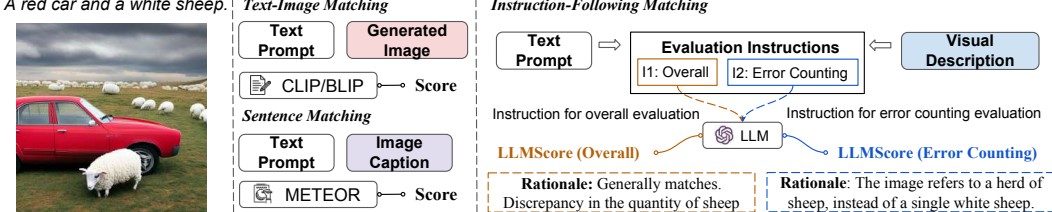

Figure 2: Comparison of Text-Image Matching, Sentence Matching, and our LLM-based Instruction-Following Matching pipeline for text-to-image synthesis evaluation. Our LLMScore automatically provides accurate scores and reasonable rationales for text-to-image synthesis based on text prompts, and visual descriptions following various evaluation instructions.

- We validate the LLMScore on a wide range of datasets (both general purpose and compositional purpose). Our proposed LLMScore achieves the highest human correlation among the commonly used metrics (CLIP, BLIP).

## 2 Background

**Existing Automatic Metrics for Text-to-Image Synthesis**  Though Inception Score (IS) [45] and the Frechet Inception Distance (FID) [18] are recognized metrics for image fidelity, they do not measure how well the synthesized images align with the text prompts. Existing metrics [21, 22, 34, 44] for measuring such alignment in text-to-image synthesis generally fall into two categories: 1) text-image matching, and 2) sentence matching. For direct comparison between the text and image, the metrics typically rely on the pre-trained text-image matching models, among these, CLIP-based and BLIP-based are most common. Alternatively, the sentence-matching pipeline transforms synthesized images into captions and then measures the sentence-level similarity with the text prompts. This can be achieved by transforming the synthesized images $v$ into free-form captions using the image captioning model BLIPv2 [25]. Then we can apply reference-based image caption metrics such as text-based CLIPScore [17] and METEOR [3] to measure the alignment.

**Large Language Models as Evaluation Metrics**  Very recently, large language models (LLMs) [32, 33] have achieved incredible performance in evaluating natural language generation tasks [14, 29, 55]. This success stems from the LLMs' powerful reasoning and instruction-following, enabling the evaluation of diverse objectives simply by altering the prompt. In addition to language-only tasks, recent studies have demonstrated the effectiveness of eliciting vision and language reasoning abilities of LLMs by incorporating image descriptions [13, 30, 40, 49] or fusing multimodal features [31, 56]. A concurrent work TIFA [19] utilizes LLMs to generate questions for validating text-to-image faithfulness. To our best knowledge, we are the first to introduce object-centric descriptions of images into LLMs for evaluating the multi-granularity compositionality in text-to-image synthesis.

## 3 LLMScore

Our proposed LLMScore aims to evaluate the alignment between the generated image and the text prompt while capturing multi-granularity compositionality. As depicted in Figure 3, LLMScore has two main components: 1) LLMs As Multi-Granularity Visual Descriptor: transforming the image into multi-granularity object-centric descriptions (Section 3.1), and 2) LLMs As Text-to-Image Evaluator: feeding the evaluation instructions into LLMs, generating the score and rationale (Section 3.2).

### 3.1 LLMs As Multi-Granularity Visual Descriptor

As illustrated in Figure 3, we first decompose the image into two-level visual descriptions: image-level global description; and region-level local descriptions. Then we employ the LLMs to fuse them into a coherent, object-centric, multi-granularity description of the image.

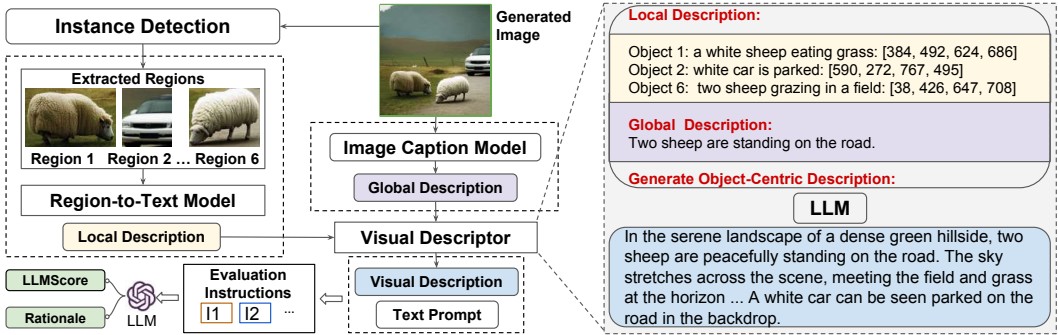

Figure 3: LLMScore pipeline for Text-to-Image evaluation. This image is generated by Stable-Diffusion-2 using the text prompt "A red car and a white sheep.", which is sampled from the Concept Conjunction dataset. I1 and I2 represent two different evaluation instruction settings: 1) Overall and 2) Error Counting. Each produced LLMScore is accompanied by a rationale.

**Global Image Description**  The global image description is composed of image-level captions and image meta-information. Given an image, for example, in Figure 3, we first transform it into a global description using the state-of-the-art image captioning model BLIPv2, which encapsulates the primary context of the image in a single sentence. Then we concatenate it with the meta-information of the image, such as the resolution (i.e. 512×512), which aids in the understanding of the location of each object, and interaction among objects.

**Local Region Descriptions**  Global description can only offer high-level information for the image, detailed descriptions for each region are necessary to capture object-level local information. Here we utilize GRiT [50] to extract regions of interest and transform them into textual descriptions of the regions. GRiT is a model pre-trained with detection and dense caption objectives jointly on the Visual Genome dataset, which contains local fine-grained descriptions for objects in the image. Specifically, we format the description of each object as "`[Object]:[Dense Caption]:[Bounding box]`". As shown in Figure 3, the descriptions of all the objects are concatenated together as our local region descriptions for each image.

**Object-Centric Visual Description**  Though local region descriptions capture dense information about objects in the image, they lose global context compared with global image descriptions, which may lack accurate interpretation of spatial and interaction relationships among objects. By incorporating both the global image description and the local region descriptions, we are able to obtain object-centric visual descriptions that capture multi-granularity compositionality, such as the attributes of the objects and relationships among the objects. Specifically, we feed the local and global descriptions into LLMs (GPT-4 [32] by default) with the template "`[GLOBAL DESCRIPTION] [LOCAL DESCRIPTION] DESCRIPTION INSTRUCTION`". We fill the slot `[GLOBAL DESCRIPTION]` with the global image description and the slot `[LOCAL DESCRIPTION]` with the local region description. And the hand-crafted slot `DESCRIPTION INSTRUCTION` ("generate object-centric description" in Figure 3) is replaced with "Based on the above information of the image, generate the object-centric visual description regarding the numerical counting, shape, color, size, location, materials of the object and the spatial and interaction relationships among the objects."

## 3.2  LLMs As Text-to-Image Evaluator

Large language models have demonstrated strong reasoning abilities (mathematical, coding etc.) on many complex tasks [6]. Here we employ the LLMs to measure the alignments between the generated image and text, and utilize their reasoning ability to understand the compositional attributes of objects and the complex interactions among multiple objects. Given the above-generated visual descriptions (Section 3.1), the whole evaluation process contains three steps: 1) instruction-following rating, 2) rule-enhanced rating, and 3) rationale generation.

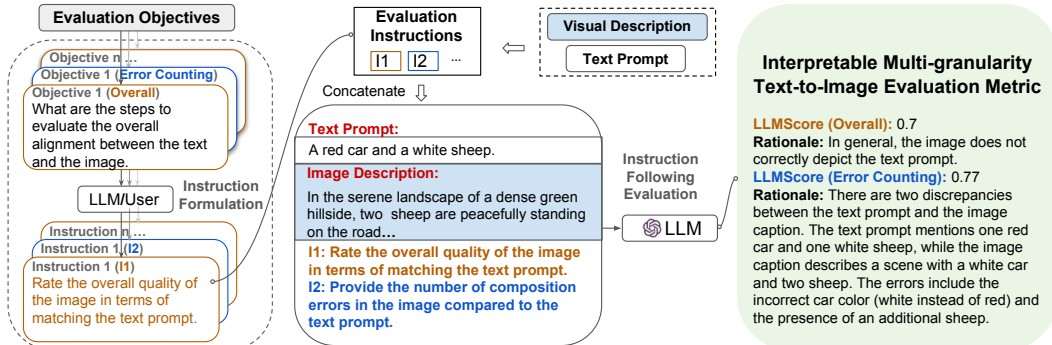

Figure 4: Large Language Models As Text-to-Image Evaluator. By defining evaluation instructions for various evaluation objectives, large language models can follow the instructions to evaluate text-image alignment based on the aforementioned visual descriptions and text prompts.

**Instruction-Following Rating**    As shown in Figure 4, we design evaluation instructions to guide the LLM in evaluating the image based on specific criteria, such as overall semantics, error counting, etc. The instructions are explicit and detailed, and can include specific guidance on error types (i.e. counting/shape/color/size) commonly seen in text-to-image models. This ensures the LLM's evaluation remains deterministic and is not influenced by any inherent biases in the pre-training data. The visual descriptions and the evaluation instructions are fed into the LLM to generate a score. For example, for *Overall* quality evaluation, the hand-crafted slot [EVALUATION INSTRUCTION] (denoted as Instruction I1/I2 in Figure 4) is replaced with "Rate the overall quality of the image in terms of matching the text prompt." Similarly, for *Error Counting* evaluation, the hand-crafted slot [EVALUATION INSTRUCTION] is replaced with "Provide the number of compositional errors in the image compared to the text prompt." We define the error type as the object-level difference. For example, the counting/shape/color/size difference in the image and text prompt should be counted as one error. This approach can either speed up the process of human annotators or serve as an interpretable and consistent evaluation pipeline.

**Rule-Enhanced Rating**    While directly using LLMs to produce evaluation scores has its merits, it presents certain challenges. One notable issue for LLM is that it can only generate discrete outputs, making it challenging to produce decimal scores even when the text prompt explicitly specifies that. To mitigate this issue, we can restrict the LLMs to rate over a wider range of scale (on a scale of 1-N, where N is default set as 100) in integer and downscale to a smaller range of scale (divided by N by default) to enforce decimal. This approach provides more flexibility to capture small discrepancies compared to directly prompting with the decimal score.

To further enhance the consistency of LLMs' ratings, we propose breaking down the evaluation process into deterministic atomic tasks and employing basic heuristic rules to imitate the human evaluation process more accurately and consistently. In the first step, the LLMs receive the concatenated information (text prompt, image description, and evaluation instruction) and produce predictions for a pre-defined sequence of atomic tasks. The atomic tasks include obtaining: 1) the number of objects specified in the text prompt (X1), 2) the number of matched objects in the image (X2), 3) the number of specified attributes in the text prompt (Y1), and 4) the number of matched attributes in the text prompt (Y2). The results for these atomic tasks are more deterministic compared with high-level tasks. In the second step, we use basic heuristics rules inspired by human evaluation processes, which reason how well the objects specified in the text prompt are presented in the image descriptions and how correctly the attributes are depicted in the image. Thus we combine the results from atomic tasks and derive the final evaluation score by $(X2/X1)/2 + (Y2/Y1)/2$. This is inspired by recent work [7, 15] that separates the calculation in LLMs to achieve more reliable results. This approach allows us to overcome the limitations of decimal value generation and consistency in LLMs. Thus we obtain evaluation results that are more accurate, interpretable, and closer to human judgment.

**Generating Rationale**    The score generation process involves the LLMs' understanding of various evaluation instructions, such as assessing the overall text-image alignment or precisely counting the number of errors in the image. Then the LLMs apply the understanding of the evaluation instruction

to the concatenated visual description and text prompt, generating a score that reflects the alignment between the image and the text at multi-granularity compositionality (image-level and object-level). Each score is accompanied by a rationale. To be specific, we add the explanation instruction prompt for LLMs, such as "Explain the overall rating X within one paragraph." for the overall objective and "Explain the error counting within one paragraph" for the error counting objective. As shown in Figure 4, our LLMScore generate reasonable rationales for both overall and error counting objectives. The rationales for the scores further provide insights into the LLM's decision-making process.

## 4 Experiments

### 4.1 Experiments Setup

**Datasets**   For general purpose evaluation (GeneralBench), we sample 200 text prompts from each dataset, including MSCOCO [27] 2014 and 2017, DrawBench [42], and PaintSkills [8]. For compositional purpose evaluation (CompBench), we sample 200 text prompts from Concept Conjunction (CC) and Attribute Binding Contrast (ABC) datasets that are designed for evaluating compositional text-to-image synthesis [12]. In total, we gather 1200 text prompts for human correlation experiments.

**Text-to-Image Models**   For each sampled text prompt, we generate two images using two widely used text-to-image models, Stable Diffusion [38] and DALL·E[37], which demonstrate extraordinary generation quality. To be specific, we use Stable Diffusion 2.1-v from Hugging Face, and DALL·E 2 using OpenAI API in April 2023. All the images are generated at a resolution of $512 \times 512$. The total text-image pairs prompts used in human correlation experiments are 2400.

**Baseline Metrics**   We consider these publicly available model-based evaluation metrics, which fall into the text-image matching pipeline (depicted in Figure 2) in evaluating text-to-image synthesis.

1. CLIP [17, 36] measures the cosine similarity of image and text prompt representations extracted from the CLIP feature extractors. This is a widely used model-based metric to measure the text-image alignment in text-to-image synthesis.

2. NegCLIP [51] uses a fine-tuned CLIP with improved compositionality understanding to measure the cosine similarity of the image and the text prompt.

3. BLIP-ITC [24, 25] uses a cosine similarity function over the extracted image and text features by BLIPv2, similarly as CLIP.

4. BLIP-ITM [24, 25] uses cross-attention to fuse multimodal features extracted by BLIPv2 to compute fine-grained similarity.

We discuss the sentence-matching pipeline as our ablations in Section 4.3. Notice that we focus on evaluating how well the synthesized images are aligned with the text prompts. Thus the widely used Inception Score (IS) and the Frechet Inception Distance (FID) for evaluating image quality are not considered in our baselines, since they do not compare the image with the text prompts.

**Implementation Details**   We extract the global image description using the pre-trained 2.7B BLIPv2 [24] model equipped with large language model OPT [54]. We extract local region description using the dense caption model GRiT [50] with ViT-Base [10] pre-trained on COCO 2017. We obtain the object-centric visual description using GPT-4 [32] as default language models to combine the global and local information. The is by default generated by GPT-4 with The GPT-4 model is the default descriptor to combine global and local descriptions into object-centric visual descriptions and the default evaluator to produce the score with rationale. All experiments conducted with GPT models are using OpenAI API from April 2023 to May 2023 with temperature 0.7 using greedy decoding by default.

### 4.2 Human Ratings

For each generated text-image pair, we ask 2 human annotators to provide ratings over these synthesized images in terms of `Overall` quality and `Error Counting`:

Table 1: Composition-focused Prompt Bench. The correlation between automatic evaluation metrics and human rankings on text-to-image synthesis. LLMScore significantly surpasses existing metrics in terms of Kendall's $\tau$ and Spearmanr's $\rho$ with $p < 0.001$.

| Human | Metric | Concept Conjunction | | | | Attribute Binding Contrast | | | |
|---|---|---|---|---|---|---|---|---|---|
| | | Stable Diffusion | | DALLE | | Stable Diffusion | | DALLE | |
| | | $\tau(\uparrow)$ | $\rho(\uparrow)$ | $\tau(\uparrow)$ | $\rho(\uparrow)$ | $\tau(\uparrow)$ | $\rho(\uparrow)$ | $\tau(\uparrow)$ | $\rho(\uparrow)$ |
| Overall | CLIP | 0.1698 | 0.2459 | −0.0049 | −0.0058 | 0.0186 | 0.0320 | 0.0396 | 0.0548 |
| | NegCLIP | 0.1724 | 0.2504 | 0.0682 | 0.0995 | 0.0151 | 0.0211 | 0.1145 | 0.1634 |
| | BLIP-ITM | 0.4058 | 0.5618 | 0.3768 | 0.5266 | 0.1799 | 0.2559 | 0.1500 | 0.2134 |
| | BLIP-ITC | 0.2378 | 0.3398 | 0.0991 | 0.1413 | 0.1982 | 0.2814 | 0.0252 | 0.0344 |
| | LLMScore | **0.4871** | **0.6956** | **0.5167** | **0.7230** | **0.4005** | **0.5480** | **0.3955** | **0.5506** |
| Error Counting | CLIP | 0.2012 | 0.2864 | −0.0782 | −0.1107 | 0.0061 | 0.0071 | 0.0914 | 0.1286 |
| | NegCLIP | 0.2245 | 0.3240 | −0.0353 | −0.0502 | −0.0339 | −0.0418 | 0.0796 | 0.1130 |
| | BLIP-ITM | 0.3341 | 0.4561 | 0.1105 | 0.1668 | 0.0696 | 0.0968 | 0.1249 | 0.1783 |
| | BLIP-ITC | 0.2210 | 0.3124 | −0.0755 | −0.1071 | 0.0895 | 0.1315 | 0.0533 | 0.0786 |
| | LLMScore | **0.3779** | **0.5443** | **0.2880** | **0.4428** | **0.1863** | **0.2821** | **0.2326** | **0.3351** |

Table 2: The correlation between automatic evaluation metrics and human rankings on text-to-image synthesis. LLMScore significantly surpass existing metrics in terms of Kendall's $\tau$ and Spearmanr's $\rho$ with $p < 0.001$.

| Human | Metric | COCO2014 | | COCO2017 | | DrawBench | | PaintSkills | |
|---|---|---|---|---|---|---|---|---|---|
| | | $\tau(\uparrow)$ | $\rho(\uparrow)$ | $\tau(\uparrow)$ | $\rho(\uparrow)$ | $\tau(\uparrow)$ | $\rho(\uparrow)$ | $\tau(\uparrow)$ | $\rho(\uparrow)$ |
| Overall | CLIP | 0.1971 | 0.2655 | 0.2227 | 0.2771 | 0.1530 | 0.2143 | 0.4715 | 0.5869 |
| | NegCLIP | 0.2164 | 0.2905 | 0.2793 | 0.3523 | 0.1463 | 0.1999 | 0.4911 | 0.6313 |
| | BLIP-ITM | 0.3252 | 0.4255 | 0.0928 | 0.1155 | 0.1044 | 0.1455 | 0.4755 | 0.6214 |
| | BLIP-ITC | 0.3465 | 0.4535 | 0.1703 | 0.2121 | 0.1569 | 0.2171 | 0.4743 | 0.5864 |
| | LLMScore | **0.3629** | **0.4612** | **0.3357** | **0.4275** | **0.2230** | **0.3023** | **0.5600** | **0.6853** |
| Error Counting | CLIP | 0.1464 | 0.2142 | 0.1888 | 0.2677 | 0.1360 | 0.1910 | 0.3052 | 0.2891 |
| | NegCLIP | 0.2116 | 0.3061 | 0.1795 | 0.2581 | 0.1179 | 0.1596 | 0.4563 | 0.4908 |
| | BLIP-ITM | 0.2251 | 0.3289 | 0.1137 | 0.1635 | 0.0871 | 0.1189 | 0.4622 | 0.4997 |
| | BLIP-ITC | 0.2636 | 0.3739 | 0.1849 | 0.2620 | 0.1506 | 0.2029 | 0.6178 | 0.6511 |
| | LLMScore | **0.2830** | **0.3992** | **0.2038** | **0.3027** | **0.2134** | **0.2865** | **0.6437** | **0.7325** |

- `Overall`: a general-purpose text-to-image evaluation, which applies to most existing metrics. Human annotators are required to rate the overall quality of the synthesized images in terms of matching the Text Prompt.
- `Error Counting`: Human annotators are required to provide the number of compositional errors in the synthesized images compared to the text prompt. The error types include: 1) compositional errors: wrong attributes (color, spatial position, shape, size, material) of the objects and wrong relationship among objects, 2) missing object errors: the objects mentioned in the text prompt are not present in the image, and 3) over-specification errors: the image hallucinates irrelevant objects in the image that are not specified in the text prompt.

The averaged inter-rater agreement is 0.62 under Krippendorff's alpha agreement measure. We provide clear guidance for human annotators, with details shown in Appendix C.

### 4.3 Human Correlation

In Table 1 and Table 2, we use Kendall's tau ($\tau$) and Spearman's rho ($\rho$) to measure the ranking correlation to both `Overall` and `Error Counting` human rating for compositional bench and general bench. All the model-based metric scores and human ratings are normalized to 0-1 for comparison.

**Overall Results** As concluded in Table 1 & 2 and Figure 5, 1). the most popular text-image alignment metric (CLIP) for text-to-image evaluations is less correlated with human ratings than

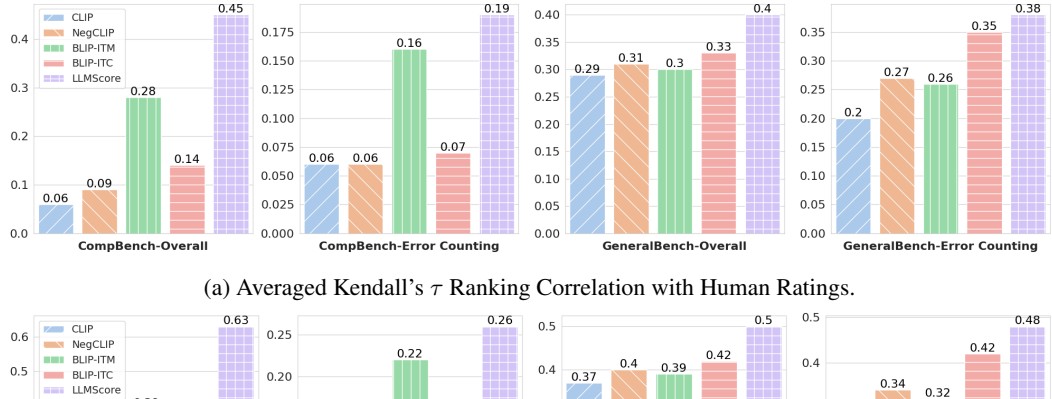

(a) Averaged Kendall's $\tau$ Ranking Correlation with Human Ratings.

(b) Averaged Spearmanr's $\rho$ Ranking Correlation with Human Ratings.

Figure 5: The rank correlation is aggregated across the compositional prompt dataset (Concept Conjunction, Attribute Binding Contrast) on the left two columns (CompBench) and the general prompt dataset (MSCOCO, DrawBench, PaintSkills) on the right two columns (GeneralBench).

expected. NegCLIP utilizes hard negatives to improve the compositionality of CLIP; 2). BLIP-based metrics (BLIP-ITM and BLIP-ITC) surpass these CLIP-based metrics (CLIP and NegCLIP). We suppose BLIP-based metrics benefits from the better object-level vision-language presentations learned by grounding tasks, compared with image-level matching learned in CLIP. 3). LLMScore is significantly better than the existing metrics with large margins.

**Accurate Error Counting in the Image is Challenging**  As shown in Figure 5, all the metrics achieve better correlation with `Overall` human ratings than `Error Counting` human ratings given the fact that there are various error types and capture each of them requires more accurate compositional visio-linguistic understanding and generation.

**Object-centric Visual Descriptions Improve Compositional Understanding**  In Figure 5, we show that our LLMScore achieves larger correlation gain over the text prompts on the compositional bench (e.g., counting, position, size, color, relations). This further confirms our superiority in capturing compositionality with the introduced object-centric visual descriptions.

**Image Captions v.s. Visual Descriptions**  We consider two categories of variants for LLM-Score:

1. Text-Caption Matching: CapCLIP and Cap-METEOR use the CLIP and METEOR to measure the similarity between the captions (which is the global image description in Section 3.1) of the synthesized images and text prompts.

2. Text-Description Matching: DescCLIP and DescMeteor, which use the CLIPScore [17] and METEOR [3] to calculate the similarity score between visual descriptions (in Section 3.1) and text prompts.

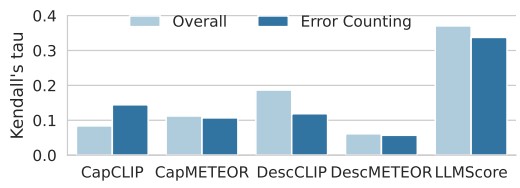

Figure 6: Comparison between Sentence Matching (CLIP, METEOR) and Instruction-Following Matching (LLMScore). LLMScore achieves the best averaged Kendall's $\tau$ correlation with human ratings over the GeneralBench (MSCOCO, Draw-Becnh, and PaintSkills).

Table 3: Effects of Large Language Models. LLMScore (Overall) can obtain performance gain from GPT-4 compared with using GPT-3.5 as the evaluator. Using GPT-4 as default visual descriptors do not improve the evaluation performance when only using image caption metrics (DescCLIP, DescMeteor). All numbers are averaged on the GeneralBench (MSCOCO, DrawBecnh, and PaintSkills).

| Human | LLM | DescCLIP | | DescMeteor | | LLMScore | |
|-------|-----|----------|----------|------------|----------|----------|----------|
| | | $\tau(\uparrow)$ | $\rho(\uparrow)$ | $\tau(\uparrow)$ | $\rho(\uparrow)$ | $\tau(\uparrow)$ | $\rho(\uparrow)$ |
| Overall | GPT-3.5 | 0.1479 | 0.1956 | 0.0042 | 0.0073 | 0.2480 | 0.3285 |
| | GPT-4 | 0.1128 | 0.1485 | 0.0297 | 0.0374 | 0.2793 | 0.3649 |
| Error Counting | GPT-3.5 | 0.0467 | 0.0670 | −0.0597 | −0.0835 | 0.2205 | 0.3013 |
| | GPT-4 | 0.0149 | 0.0228 | −0.1087 | −0.1494 | 0.2131 | 0.2981 |

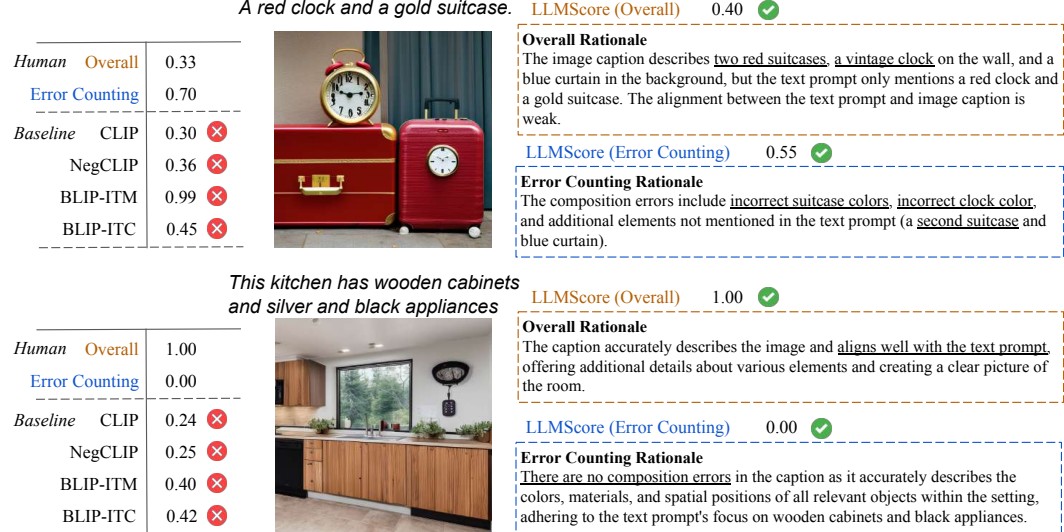

Figure 7: Examples showing the LLMScore captures the object-level discrepancies (*TOP*) and similarities (*BOTTOM*) between the image and the text prompt. The two text prompts are sampled from the Concept Conjunction dataset.

The main drawback of such a sentence-matching pipeline is that the caption metrics favor the coverage or similar language structure instead of capturing compositional semantics. Figure 6 shows the average correlation (Kendall's $\tau$) on the general bench set, LLMScore significantly outperforms these two variants, indicating that, caption metrics is not a good means to measure the alignment between image and text. Despite the advance introduced by object-centric visual descriptions, the limitations associated with using caption metrics to measure semantic similarity hinder the performance gain of text-description matching metrics DescCLIP and DescMETEOR over text-caption matching metrics CapCLIP and CapMETEOR.

**Differing Large Language Models** In Table 3, we compare two variants of large language models: GPT-3.5 and GPT-4. 1). For the same group (for example, GPT-4 based) visual descriptions, LLMScore is better than DescCLIP and DescMeteor, indicating that LLMs reasoning is the key component to capture the semantic relations in the visual descriptions. 2). We observe that GPT-4 based LLMScore (Overall) has an average better correlation than GPT-3.5, indicating the performance benefits from the better reasoning ability in the larger-scale language model. However, GPT-4 based LLMScore (Error Counting) is only comparable with GPT-3.5, indicating that the counting is still un-resolved in large-scale language models.

### 4.4 Multi-Granularity Rationale for Text-to-Image Evaluation

In Figure 7, we show two examples, our LLMScore not only produces human-correlated text-image scores but also provides a rationale for each metric value. The rationale correctly explains both the differences and similarities between the descriptions and text prompts. The model performs well in explaining the errors counted and the overall comparisons.

## 5  Conclusion

In this paper, we re-examine the existing model-based text-to-image metrics and propose LLMScore to evaluate text-to-image synthesis by unveiling the power of large language models. Our LLMScore can capture the multi-granularity compositionality between the synthesized images and the text prompt, producing accurate alignment scores with rationales. Our LLMScore demonstrates significantly better correlation with human scores on several datasets, paving the way for a more adaptable text-to-image evaluation, capable of following human instructions to evaluate the text-image alignment.

## Broader Impact

The framework proposed in this paper first integrates GPT-4 for text-to-image evaluation and showcases how to take advantage of the existing large-scale pre-trained models (GPT-4) for measuring the alignment between the generated images and text, we also propose a new metric, LLMScore which provides interpretable rating and well aligns with the human scores on several datasets. This work sheds light on the value of large language models on the evaluation of text-to-image synthesis, we hope it can help the future text-to-image synthesis work on improving the groundedness and compositionality, either as a reward signal or evaluation metric; our preliminary work on the interpretability of LLMScore, may have the potential to be used for explanation, controllable generation, and image editing.

## Limitations

One limitation of our work is that it relies on GPT, which is not free for the public, and may limit its fast plug-in capability, future work may consider replacing this component with a publicly available LLM model (e.g., LLaMA) or our in-house finetuned image captioning model. Another potential issue for this work is, since it incorporates the exsiting large language models, it may inherit its own biases that could propagate to the metric. The future work who considers adopting our LLMScore metric should be cautious on the specific domains to make sure no harmful biases get propagated.

## Acknowledgments

The research was sponsored by the National Science Foundation under Grant No. 2048122. The views and conclusions contained in this document are those of the authors and should not be interpreted as representing the official policies, either expressed or implied, of the sponsor.

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

# A Background

## A.1 Text-conditioned Image Synthesis

The popular large-scale image generation models such as Imagen [43], DALL-E2 [37] and Stable Diffusion [39] demonstrate extraordinary generation quality, but a subtle difference in input prompt could lead to a dramatic change of semantic style, thus not directly suitable for image editing [2, 41]. Besides, the prompt-to-prompt [16] achieves image editing with cross-attention control on the observation of interaction between the pixels to the text embedding. InstructPix2Pix [4] leverages the complementary abilities of a pre-trained language model GPT-3 [5], and a pre-trained text-to-image model Stable Diffusion [38] to generate large pairs of multi-modal training data and perform image editing following human instructions. ControlNet [52] learns task-specific conditions in an end-to-end way and achieves robust control effects. Recently, growing interests [35] in computer vision focus on aligning text-to-image synthesis [23] or visual editing [53] by using human feedback. Typically, a reward model is expected to evaluate images by training on task rewards using proximal policy optimization (PPO [46]).

## A.2 Image-level Evaluation Metric

**CLIP-based**   CLIP [36] was pre-trained on large-scale image-caption pairs through contrastive learning, making it a highly versatile tool for natural language processing and computer vision applications. CLIPScore [17] measures the cosine similarity value of image and caption representations extracted from the CLIP feature extractors. Formally, $CLIPScore = max(cos(f_I(v_i), f_C(c_i)), 0)$, where $f_I, f_C$ is the image and caption feature extractor. CLIPScore is a reference-free metric and outperforms previous reference-based metrics like CIDEr [48] and SPICE [1]. Within the text-to-image domain, previous work also relies on the same approach to measure the alignment between the text prompt and the generated image. However, vision-and-language models exhibit deficiencies in compositional understanding and are insensitive to word orders. In [51], they show BLIP [25] and CLIP [36] only achieve random chance level understanding ability on attribution, relation, and order understanding. They furthermore propose NegCLIP to improve the original CLIP model by generating additional hard negative captions and optimizing the same contrastive objective. They show that obtaining specific and low-cost negative examples can result in significant enhancements in compositional tasks without losing existing ability.

**BLIP-based**   BLIP [25] filters out noisy synthetic captions to effectively make use of the noisy web data through bootstrapping based on a novel multimodal mixture of Encoder-Decoder. Beyond that, BLIPv2 [24] introduces Query Transformer that bootstraps image and text representation learning and then bootstraps large language model for image-to-text generations. BLIPv2 achieves state-of-the-art performance on a wide range of understanding-based and generation-based vision-language tasks, including image-text retrieval, image captioning, and visual question answering. We suppose the grounding objective in BLIPv2's pre-training can bootstrap its performance in evaluating text-to-image synthesis. Specifically, we utilize BLIPv2 to compute the image-text matching score using "ITM" head and "ITC" head. BLIP-ITC uses a simple cosine similarity function over the extracted image and text features. In contrast, BLIP-ITM uses cross-attention to fuse multimodal features to capture fine-grained similarity.

## A.3 Evaluation Datasets

The MSCOCO [27] has been widely used for object segmentation, although there is a dearth of varied prompts, indicating a lack of diversity. Winoground [47] is designed for evaluating the ability of vision and language models to conduct visio-linguistic compositional reasoning. For a pair of two distinct images, their captions are composed of identical sets of words, but in a different order. Many state-of-the-art vision and language models only achieve random chance performance, making it a good testbed for evaluation. DrawBench [42] tackles prompt diversity issues by collecting challenging descriptions for image generation. There are a set of 11 prompt categories that test various capabilities of models, including the ability to accurately depict colors, numbers of objects, spatial relations, and text, as well as more complex prompts such as long textual descriptions, rare words, and misspelled

prompts. In [8], they evaluate the visual reasoning of text-to-image models and propose PaintSkills, a diagnostic dataset and evaluation toolkit designed to measure object recognition, counting, and spatial understanding. Recent studies in compositional text-to-image synthesis[12] collect Concept Conjunction prompt dataset which focuses on two objects with different colors in the text prompt, and Attribute Binding prompt dataset that is sampled from COCO captions.

## B    More Results

In Table B, we show the full table that includes variants of our LLMScore (CapCLIP, CapMETEOR, DescCLIP, DescMETEOR) on General Bench.

Table 4: The correlation between automatic evaluation metrics and human rankings on text-to-image synthesis. Our devised metrics LLMScore significantly surpass existing metrics in terms of Kendall's $\tau$ and Spearmanr's $\rho$ with $p < 0.001$.

| Human | Metric | COCO2014 | | COCO2017 | | DrawBench | | PaintSkills | |
|---|---|---|---|---|---|---|---|---|---|
| | | $\tau(\uparrow)$ | $\rho(\uparrow)$ | $\tau(\uparrow)$ | $\rho(\uparrow)$ | $\tau(\uparrow)$ | $\rho(\uparrow)$ | $\tau(\uparrow)$ | $\rho(\uparrow)$ |
| Overall | CLIP | 0.1971 | 0.2655 | 0.2227 | 0.2771 | 0.1530 | 0.2143 | 0.4715 | 0.5869 |
| | NegCLIP | 0.2164 | 0.2905 | 0.2793 | 0.3523 | 0.1463 | 0.1999 | 0.4911 | 0.6313 |
| | BLIP-ITM | 0.3252 | 0.4255 | 0.0928 | 0.1155 | 0.1044 | 0.1455 | 0.4755 | 0.6214 |
| | BLIP-ITC | 0.3465 | 0.4535 | 0.1703 | 0.2121 | 0.1569 | 0.2171 | 0.4743 | 0.5864 |
| | CapCLIP | 0.0263 | 0.0335 | -0.0274 | -0.0315 | 0.0056 | 0.0072 | 0.3035 | 0.3751 |
| | CapMETEOR | 0.0710 | 0.0960 | 0.0512 | 0.0650 | 0.0951 | 0.1312 | 0.2315 | 0.3056 |
| | DescCLIP | 0.1377 | 0.1799 | 0.1130 | 0.1424 | 0.1136 | 0.1557 | 0.3825 | 0.4917 |
| | DescMETEOR | 0.1175 | 0.1549 | 0.0567 | 0.0702 | 0.0028 | 0.0048 | 0.0678 | 0.0827 |
| | **LLMScore** | **0.3629** | **0.4612** | **0.3357** | **0.4275** | **0.2230** | **0.3023** | **0.5600** | **0.6853** |
| Error Counting | CLIP | 0.1464 | 0.2142 | 0.1888 | 0.2677 | 0.1360 | 0.1910 | 0.3052 | 0.2891 |
| | NegCLIP | 0.2116 | 0.3061 | 0.1795 | 0.2581 | 0.1179 | 0.1596 | 0.4563 | 0.4908 |
| | BLIP-ITM | 0.2251 | 0.3289 | 0.1137 | 0.1635 | 0.0871 | 0.1189 | 0.4622 | 0.4997 |
| | BLIP-ITC | 0.2636 | 0.3739 | 0.1849 | 0.2620 | 0.1506 | 0.2029 | 0.6178 | 0.6511 |
| | CapCLIP | 0.0266 | 0.0362 | -0.0068 | -0.0085 | 0.0544 | 0.0704 | 0.4963 | 0.5332 |
| | CapMETEOR | 0.0822 | 0.1197 | 0.0004 | 0.0013 | 0.0173 | 0.0192 | 0.3274 | 0.3636 |
| | DescCLIP | 0.1433 | 0.2145 | 0.0338 | 0.0477 | -0.0039 | -0.0022 | 0.2978 | 0.3289 |
| | DescMETEOR | 0.1398 | 0.2010 | -0.0829 | -0.1198 | -0.1348 | -0.1791 | 0.0881 | 0.0924 |
| | **LLMScore** | **0.2792** | **0.4006** | **0.2138** | **0.3125** | **0.2125** | **0.2839** | **0.6444** | **0.7066** |

## C    Human Annotation

For each image-text pair, we ask 2 annotators to rate the overall and error counting. We will show the details of annotation interface in Section C.1.

### C.1    Human Ratings Interface

In Figure 8, we show the interface for human ratings over the image quality from two objectives, overall and error counting. Human annotators are required to rate the overall quality of the image on a scale of 1-10 and count the errors in the image on a scale of 0-9.

**Question**

Given the Text Prompt **"A gold chair and a red clock."**:

**[Overall Quality]**According to the Text Prompt, verify the Overall and Compositional quality of the Generated Images below by rating on a scale of 10.

Text Prompt **"A gold chair and a red clock."**
Rate the overall quality of the Generated Images in terms of matching the Text Prompt:

- ○ 1 - Poor.    ○ 2 - Very Bad.    ○ 3 - Bad: Low quality, merely aligned with the text prompt.    ○ 4 - Not okay.

- ○ 5 - Neutral (Leaning Negative)    ○ 6 - Neutral (Leaning Positive)    ○ 7 - Okay

- ○ 8 - Good: High quality, aligned with the text prompt.    ○ 9 - Very Good    ○ 10 - Perfect.

Given the Text Prompt **"A gold chair and a red clock."**:

Text Prompt        **" gold chair and a red clock."**
 **[Error Counting]**   Provide the number of composition errors Y (scale: 0-9) in the    Generated Images    compared to the Text Prompt. One error should be counted for each incorrect color, spatial position, shape, size, material, or relationship among objects. If an object category mentioned in the Text Prompt is missing in the caption, count it as 4 errors. The over-specifications in the image caption should be counted as only one error::

- ○ No composition error found.    ○ 1 composition error.    ○ 2 composition error.    ○ 3 composition error.

- ○ 4 composition error.    ○ 5 composition error.    ○ 6 composition error.    ○ 7 composition error.

- ○ 8 composition error.    ○ 9 or above 9 composition error.

Figure 8: Amazon Mechanical Turk Platform. Questions Layout for Human Raters for Overall and Compositional ratings of the generated image given the text prompt.

# D    Visual Descriptions

Our approach involves utilizing both global and local descriptions of an image. Initially, a general caption is generated for the image. Then, a dense caption model is employed to describe the objects in detail. This technique enables the extraction of both the overall context of the image and the specific attributes of individual objects, thereby providing a comprehensive description of the image.

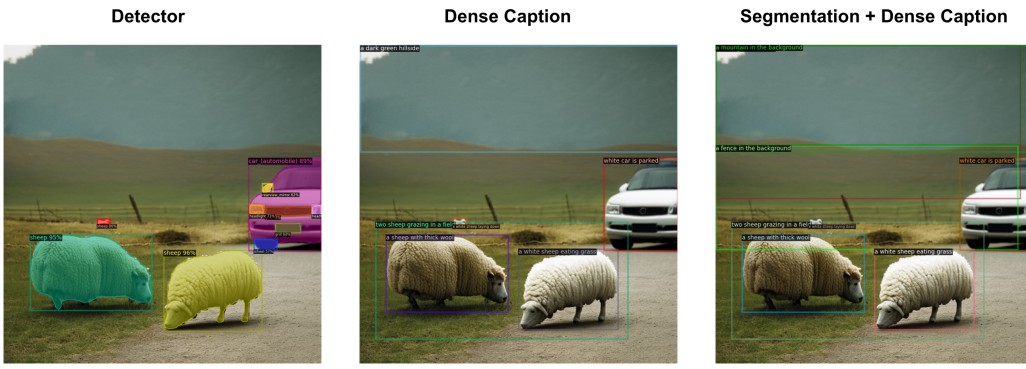

Figure 9: Comparison between, detector, dense caption, and segmentation results over the synthesized image. Alternatively, an object segmentation model can be applied to identify the objects in the image more accurately.

# E    Example of GPT Text-to-Image Evaluation.

| Stage | Example |
|-------|---------|
| Prompt | a light brown bathroom with a white sink and mirror. |
| Description | This image showcases a clean and modern bathroom with a white theme. The main focus is the white sink, which is placed on a wooden bathroom counter. The counter is brown and extends across the entire width of the image, covering the lower part of the white wall behind it. Above the sink, there is a large mirror on the wall that reflects the shower curtain as well as a silver light fixture with two light bulbs, which are switched on and provide ample light to the room. The silver faucet complements the white sink and is slightly towards the left side of the sink. A white soap dish is positioned on the right side of the counter, next to an additional sink. A silver handle can be seen on a drawer beneath the counter, near the sink. On the left side of the picture, a brown wooden door can be partially seen, while on the right side, a small part of a white toilet is visible. The overall setup of the bathroom, with its white walls, bright lights, and well-arranged counter, gives off a spacious and organized vibe. |
| Rating Rationale | Overall Semantics: 0.8/1.0 
 The alignment between the image caption and the text prompt is mostly good. The generated image caption describes a white sink, wooden counter, mirror, and other elements like a soap dish, faucet, and lights that complement the room. This is not far from the text prompt of a light brown bathroom with a white sink and mirror. |
| Rating Rationale | Error Counting: 1 (number of errors) 
 The composition error mainly lies in the description of the color of the bathroom. The text prompt mentions a light brown bathroom, but the image caption emphasizes a white theme, with only the wooden counter and door being brown. |

Table 5: Examples of GPT Text-to-Image Evaluation: the prompt is used for image generation, while the description is generated by GPT-4

We showcase examples for text-to-image evaluation in Table 5 and Table 6 on synthesized images in Figure 10. The Prompt section illustrates the initial input provided to the text-to-image model, which in this instance is a short description of a bathroom. The subsequent Description section is the multi-granularity object-centric description for the image. Here, the model describes a light-filled, modern bathroom with various elements like a white sink, mirror, soap dish, and a wooden counter. Then there are two rating settings, the first is *Overall* (Overall Semantics), and the second is *Error Counting*. Both settings evaluate the text-to-image synthesis with a score and a rationale. Here, a score of 0.8 out of 1.0 reflects a high level of correspondence between the synthesized image and the prompt, despite a few discrepancies. And 1 represents there are only one errors found in the synthesized image compare to the prompt.

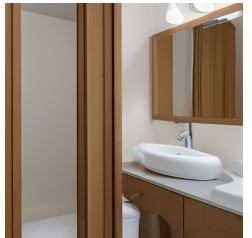 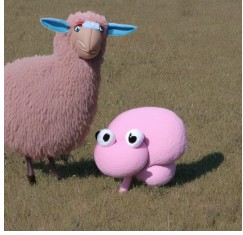

Figure 10: *Left*: Synthesized by Stable-diffusion-2 with text prompt "A light brown bathroom with a white sink and mirror" sampled from Attribute Binding Contrast dataset. *Right*: Synthesized by Stable-diffusion-2 with text prompt "A brown bird and a pink sheep." sampled from COCO.

| Stage | Example |
|-------|---------|
| Prompt | a brown bird and a pink sheep. |
| Description | In this vivid image, there are two unique pink animals standing on a large patch of green grass that covers the entire ground. On the left side, there is a pink sheep with a blue tag attached to its ear. The sheep has distinct blue ears and a prominent head, showcasing a contrast in colors against its pink body. It also has two sturdy black legs standing firmly on the grass. On the right side of the image, there is a pink pig with black eyes. The pig's eyes are embedded on a pink and white stuffed animal-like face, giving it a cute and charming look. The pink sheep and the pig are casting shadows towards the bottom of the image, blending with the green grass beneath. Overall, the colorful and playful scene provides a fantastical and captivating atmosphere, with the pink sheep and pink pig standing as the central focus. |
| Rating Rationale | Overall Semantics: 0.8/1.0
There is a clear focus on the pink sheep in the image caption, but the prompt's mention of a brown bird is entirely missing. The overall composition is good and somewhat aligned, but it could be better. |
| Rating Rationale | Error Counting: 4 (number of errors)
The composition error comes from the missing brown bird (4 errors). The color, size, and material of the pink sheep, as well as the relationship between the two animals are well-described, making this error count relatively low. |

Table 6: Examples of GPT Text-to-Image Evaluation: the prompt is used for image generation, while the description is generated by GPT-4

