# OpenReview forum: "LLMScore: Unveiling the Power of Large Language Models in Text-to-Image Synthesis Evaluation"
_NeurIPS.cc/2023/Conference — NeurIPS 2023 poster_

### Official Review · Reviewer_ZDTU · 2023-06-16

**Soundness:** 3 good
**Presentation:** 3 good
**Contribution:** 3 good
**Rating:** 6
**Confidence:** 5

**Summary:**

The authors proposed a framework to evaluate text-to-image synthesis with multi-granularity compositionality powered by LLMs. Two big steps are transforming an image into whole-level and object-level visual descriptions in a text to be used for a part of the prompt to LLMs, then LLMs measure the alignment with a detailed instruction prompt to give the LLMScore. They argue that the correlation between LLMScore and human judgments, along with qualitative evaluations with the generated rationale, confirm the superiority compared with commonly used metrics like CLIPScore.

**Strengths:**

They proposed an evaluation framework exploiting the power of LLMs to evaluate the challenging text-to-image generations. Giving the rationale for evaluation justification would be a new feature compared with previous works. The experimental procedure is solid to evaluate the proposed metric, although the comparison is limited to commonly used methods, i.e., cosine similarity or cross-attention using CLIP and BLIP features. The manuscript is well-written, well-organized, and easy to follow.

**Weaknesses:**

W1. The most concern is the comparison with recent work from NeurIPS 2022, "Mutual Information Divergence: A Unified Metric for Multimodal Generative Models" (MID) [17] (the citing number is from the submitted paper). MID similarly exploit pretrained features, i.e., CLIP, but possibly with BLIP, to calculate the text-and-image matching score; however, their method (MID) uses the statistical distribution of the features for finer evaluations than cosine similarity. The authors should compare this recent representative work to show the effectiveness and significance of the proposed method. Specifically, what would be the results of MID using BLIP features for fair comparison?

W2. Does visual descriptor have significance? GPT-4 can accept images as inputs, which means instead of visual descriptions (Sec. 3.1), one may exploit the power of LLMs directly, giving a synthesized image as a prompt. Fine-grained examination of visual input would get good results rather than using cumbersome generations of visual descriptions utilizing other pretrained models. This ablation would support the argument for the core role of the proposed visual descriptor. Otherwise, this work would be reduced to a naive application of multimodal LLMs, e.g., GPT-4.

W3. Robustness to error accumulations. LLM Evaluator inevitably relies on the accuracy of visual descriptions. Since there is no systemical analysis of the catastrophic degradation from an error in visual descriptions may hinder the reliability of the proposed method.

I am eager to reassess my score based on the authors' faithful feedback in handling the issues raised above.

**Questions:**

Figure 2-4 includes the brand logo of OpenAI. According to their brand guideline (https://openai.com/brand#things-to-avoid-when-using-our-logo), the authors should not alter their logo using custom colors. Besides, the authors used the OpenAI logo to represent LLMs; however, it is inappropriate to represent LLMs with the brand logo of a for-profit company in the manuscript.

**Limitations:**

L1. This work decisively relies on the performance of pretrained models. 1) image captioning model BLIPv2, 2) GRiT [41] for local region descriptions, and 3) GPT-4 for LLM evaluator. How do you assess the robustness toward potential errors of visual descriptors or LLM evaluators?

L2. Social bias and fairness issues can also be inherited from pretrained models.

L3. Without the commercial license to use Open API, reproducibility cannot be checked or used. Could you provide the results using the open-sourced LLM evaluators?

---

> ### Author Rebuttal · Authors · 2023-08-09
>
> Weaknesses
> 1. We thank the reviewer for pointing out this interesting work. We found out this MID metric require the real image features and fake image features with respect to the text prompt for text-to-image synthesis evaluation. This a different setting from recent text-to-image benchmark that is not paired real images, but only the text prompts (e.g., Draw Bench, ABC-6000, CC-500). And our setting is following this trend that our GeneralBench and CompBench are not all paired with reference images. We will discuss this difference in the related work.
> 2. The benefit of the visual descriptor comes from two aspects: 1) it provides the breakdown interpretability of the evaluator and it’s easy to trace any evaluation error caused by the module, 2) it’s flexible to be replaced by new SOTA submodules. Besides, the GPT-4 image input API is not released to the public, we’re still the pioneer to pushing the text-to-image synthesis evaluation by utilizing existing released large language model capability.
> 3. Thank the reviewer for this insightful suggestion. We ask annotators and manually sample from the intermediate output to check the quality of the visual descriptor and analyze how many portions of bad evaluations are caused by this. And we find out that most of the accuracy of visual descriptions regarding the synthesis evaluation satisfies the evaluation purpose. Though, it remains a bottleneck of the system and we keep it a fundamental solution and keep the modules able to be replaced with any new SOTA modules to furthermore mitigate the error propagation. We want to highlight that the benefit from this interpretable pipeline can also point out the future improvement directions in this line of research.
>
> Questions
>
> Thank you for pointing this out. We will remove the logo of OpenAI and use a new logo to represent the LLMs in the paper.
>
> Limitations
> 1. We agree that the social biases and fairness from the pre-trained models should be considered important. We’re glad to incorporate the mitigation strategies on top of these models to reduce these biases and at the same time still benefit from the rich knowledge from these models.
> 2. Thank you for pointing it out. We’ve supported the open-sourced LLM Vicuna to foster future research.

---

> > ### Comment · Reviewer_ZDTU · 2023-08-11
> > **Glad to hear the feedback**
> >
> > Weakness 1. Although you could exploit the idea of MID metric to apply without fake image features, it seems out-of-scope and less relevant. As you suggested, the discussion of the difference with this related work would be sufficient.
> >
> > I am glad to hear the feedback and cannot help but raise the score from 5 to 6, leaning toward acceptance.

---

### Official Review · Reviewer_nuAT · 2023-07-02

**Soundness:** 3 good
**Presentation:** 3 good
**Contribution:** 3 good
**Rating:** 6
**Confidence:** 4

**Summary:**

This paper presents a new method LLMScore to evaluate the alignment between a textual prompt and the generated image (especially in the context of the text-to-image diffusion models).  The key motivation behind this work is to address the inability of popular vision-language models like CLIP to deal with the fine-grained details in a complex prompt.
As a result, in this paper, vision-language models such as BLIP is used to caption the image as well as provide object-centric descriptions. These are concatenated to create a final image description using an LLM. Furthermore, the alignment between the text prompt and the image description is computed using an LLM. Experiments on a variety of text-to-image generation benchmarks indicate that the proposed method LLMScore significantly outperforms prior methods of directly measuring the similarity of image and text embeddings.

**Strengths:**

1) The problem tackled in this paper is an important one, especially due to the inadequacy of existing models.
2) The method proposed in the paper is also quite logical (since using an LLM would be able to perform better fine-grained understanding, as opposed to matching coarse grained features)
3) The experimental results achieved in the paper are quite satisfactory, and show that LLMScore is able to achieve much better results as compared to the standard measures for evaluating image-text alignment.

**Weaknesses:**

One concern about this paper is regarding the following paper [a], which could be treated as contemporary work. While it is understandable that this paper is certainly not required as a comparison as it was released less than 2 months before the submission deadline, there are a lot of similarities between [a] and LLMScore. Both rely on a vision-language model to provide descriptions of the image, and use an LLM to reason over the correctness of the generated image. Of course, there are sufficiently many differences between these 2 works ([a] first generating questions using an LLM and then answering using a VQA model, versus LLMScore generating captions/descriptions first, and then using an LLM to evaluate the description), however, it might be a good idea to atleast acknowledge [a] as a contemporary work in the related work section.

Regarding the work itself, an interesting analysis to delve into (given the fine-grained and interpretable nature of LLMScore), would be to break down the errors/inaccuracies into the errors caused due to either the perception models, or with the LLMs (either in constructing the global image description or with the evaluation). Such a breakdown would also enable future work to focus on addressing the weakest aspect of LLMScore to improve it further. Unfortunately, there is no analysis regarding the failure-cases of LLMScore which would have been extremely useful for any reader.


[a]: Hu et al. TIFA: Accurate and Interpretable Text-to-Image Faithfulness Evaluation with Question Answering, Arxiv 2023

**Questions:**

I have one question regarding the reproducibility of the method. Specifically, with GPT-4, the default temperature of 0.7 is used. In my understanding, this would mean that different runs of the LLM could give different results. Therefore, it might make sense to repeat the experiments multiple times so that the stability of the results are ensured. Of course, if I am missing something, a clarification would be very useful.


**Limitations:**

Yes, there is a limitations section in the supplementary material, which covers these issues satisfactorily.

---

> ### Author Rebuttal · Authors · 2023-08-09
>
> Weaknesses
>
> 1. Thank you for the good pointer to this great work. And yes, our work differs from this work in multiple aspects, though they share the same spirit to make the evaluation of text-to-image synthesis more interpretable and fine-grained. We will surely add this contemporary work in the related work and discuss the differences as well.
> 2. Exactly, the breakdown interpretability is one of the most appealing benefits from LLMScore. And we observe the failure-cases, mainly come out: 1) when the synthesized images are in bad quality and even human cannot recognize the objects in the images, the descriptions of the image may make hallucinations, 2) when the visual descriptions are all correct, the LLM can sometimes make a mistake in following the evaluation instruction. Potential solution would be further fine-tuning the visual descriptor in the image synthesis domain to make the descriptions more robust. And we can also instruction-tune the large language model to be more aligned with human evaluations in comparing the text prompt and visual descriptions.
>
> Questions
>
> Thank you for pointing this out. Yes, the different runs of the LLM could give different results. Thus the process of generating the visual descriptions and evaluations using LLMs is repeated twice altogether in our experiments to ensure the robustness of the performance gain over the baseline metrics.

---

> > ### Comment · Reviewer_nuAT · 2023-08-18
> >
> > Thank you for the response!

---

### Official Review · Reviewer_mWLn · 2023-07-04

**Soundness:** 3 good
**Presentation:** 3 good
**Contribution:** 3 good
**Rating:** 7
**Confidence:** 4

**Summary:**

This paper proposes a new automatic evaluation framework named LLMScore with multi-granularity compositionality to evaluate the synthesized image quality of the mainstream text-to-image synthesis models.
Its core approach is to tranform images into global and text descriptions through image captioning and region-to-text models, then follow by an LLM to as a visual descriptor and a text-to-image evaluator with rationable output.
On six image captioning benchmark datasets, the correlation between its LLMscore and human evaluation results far exceeds the existing commonly used clip-like text-image alignment indicators, reaching into a new SOTA level.
Extensive experimental analyzes strongly confirm the usability of the method.

**Strengths:**

1. I truly appreciate the novelty of this paper. It dexterously transform synthesized images into text descriptions, so that it can make full use of LLM's powerful text understanding and reasoning capabilities with a series of hand-crafted constructed Prompts (including sentence analysis, text-image alignment scores estimation, and image generation error statistics). In my standpoint, compared to the previous Clip-like's text-image matching score, this paper undoubtedly set a new T2I quality evaluation paradigm. Aslo, considering the vast influence brought by the AIGC community at present, I believe the impact this paper can bring to the community is significant.

2. The method of this paper is solid, not only in the multi-level extraction of image description, instruction-based and heuristic-based scoring mechanism with rationable generation, but also the considerable performance gains compared with multiple strong baseline methods is also a strong proof.
The selected evaluated T2I models are currently the two most popular classes - stable diffusion and DALL·E.
In addition, this paper gives ablation experiments of different LLMscore variants and LLMs (GPT3.5 vs GPT4), further confirming the necessity of each component.

3. It brings my great joy is that this work introduces its main ideas and methods, as well as experimental settings and results, etc. in an easy-to-understand manner, so that readers can easily understand.

**Weaknesses:**

Some of the following problems in my perspective may need to solve for the future deployment:

1. The scoring criteria for "Human Annotation" in the appendix should be more specific, and it should be able to guide the annotators to rate the quality of image-text alignment step by step. For example, what is the difference between a score of "4" and "5", is it because of the color generation of the object or the amount is biased?
To be honest, if the results of human annotation are not fine-grained and accurate enough, it may seriously destroy people's trust in the usability of the system.

2. The experiment and analysis can be more perfect, but considering that this article is a fundamental work, the time for urgent submission of ddl may not be enough, so this amount of completion is completely acceptable to me.

3. There are several different models applied in the overall evaluation pipeline. If a large model can be unified in the future to complete all evaluation steps (such as GPT4), the evaluation scheme may be more practical.

**Questions:**

What does the average rank correlation of the two items in Figure 5 refer to? Could the author provide a more clearly clarification and revise it in their next edition?

**Limitations:**

As the author said in the supplementray limitation part, relying on OpenAI's closed-source GPT model and the internal bias of the LLM model are obstacles faced by this proposed evaluation framework.

---

> ### Author Rebuttal · Authors · 2023-08-09
>
> Weaknesses
>
> 1. In addition to the interface visualized in Figure 8, we do provide clear comparisons to show what is a score “X” example. We agree though it might be better in the future to devise a new standard human evaluation pipeline for this.
> 2. Thank the reviewer for the suggestion. We will add more analysis of LLMScore such as what are more possible other evaluations that can be aligned with human annotators, and what are the limitations.
> 3. We agree, though LLMScore will still maintain its strengths of fine-grained interpretability of the evaluation process compared with a large foundation model.
>
> Questions
>
> As Table 1 shows the individual ranking correlation on each data source from the Composition-focused Prompt Bench (CompBench), the Figure 5 shows the averaged rank correlation of multiple data sources. The same goes to the GeneralBench (Tabel 2 vs. Figure 5).
>
> Limitations
>
> We now support the open-sourced LLM (e.g., Vicuna). We hope it will foster future research based on our work.

---

### Official Review · Reviewer_BWFw · 2023-07-06

**Soundness:** 3 good
**Presentation:** 3 good
**Contribution:** 3 good
**Rating:** 7
**Confidence:** 4

**Summary:**

This paper proposes a new evaluation metric for text-to-image models. The method works as follows:
- Prompt the Blip-2 VLM to generate an overall text description for the image.
- Use GRiT to generate coordinates and language descriptions for each object in the scene
- Use GPT4 to combine these text descriptions into an overall description of the image.
- Prompt GPT4 to score the image, both by asking it for the overall alignment and by asking it to to count the number of errors in the presence/absence or attributes of objects in the scene.
They show this score correlates better with human judgment than common baselines.

**Strengths:**

- The approach is significantly more correlated with human judgments than baselines while still being cheaper and faster than human evaluation.
- Evaluation is a big problem in text-to-image generation, and better quantitative metrics are useful.
- The paper is well-written and clear, and includes a variety of strong baselines.
- They break down the accuracy of the different types of evaluation (overall, error counting) so it's easy to see weaknesses.

**Weaknesses:**

The paper has two stages (1) getting a visual description and (2) computing a score. They explore baselines with different variations of (2)
but as far as I can tell ablate different versions of Stage 1 (i.e. using just global image descriptors vs just local object descriptors vs both while using the LLMScore scoring method).

(I think this would be a useful extension, but the paper is strong without it.)

**Questions:**

Consider mentioning similar concurrent works (e.g. TIFA https://tifa-benchmark.github.io/)



- Does the image get penalized for including objects which were not specified in the prompt but make sense in context (e.g. if it asked for a plate with silverware and a cup next to it, does the model get penalized for putting them on a table? Or for including a dining room background which includes miscellaneous objects?)

Nit:
- Partial phrase on line 218
- Line 217 "default language models" -> "the default language model"
- Table 1 caption: "Spearmanr's" -> "Spearman's"
- Specify that in the "error counting" metric, lower is better (Fig 7)
- Table 3: bold best per column.

**Limitations:**

The one mentioned limitation is that error counting is still challenging.

Not mentioned -- if future works optimize against scores on this metric, models could exploit errors in these metrics. (This is true of all automated metrics on this task though.)

---

> ### Author Rebuttal · Authors · 2023-08-09
>
> Weaknesses
>
> Thank the reviewer for this thoughtful suggestion. We provide ablations to show the advantage of using both descriptions over global caption only on CLIP and METEOR metric. This also indicates that using both can better represent the synthesized image and thus we apply this with LLMScore. As suggested, we will also add quantitative analysis for this ablation in Appendix B.
>
> Questions
>
> 1. Thank you for the question. This remains an open-ended question, as some research work may take this as hallucinations. In our work, we follow the standard that is provided to the human annotators, which require the image to be strictly aligned with the text prompt. However, regarding the background scene (e.g., dining room), this will not be penalized, as both the human and model will think the misc objects as a part of the scene.
> 2. Thank you for pointing out these. We will fix these typos as suggested.
>
> Limitations
>
> 1. Yes, the error counting is still challenging. A possible solution is to break down the evaluation protocol as atomic as possible, and combine together to bypass the limitation of current models.
> 2. To mitigate this, our LLMScore supports incorporating as many submodules/evaluation instructions as possible, and thus easily to validate the robustness of the models (e.g., come out of an unseen and reasonable evaluation criteria). And the LLMScore is extensible to combine multiple off-the-shelf submodules to increase the diversity of the evaluator to avoid such short-cut attacks.

---

> > ### Comment · Reviewer_BWFw · 2023-08-16
> > **Thanks**
> >
> > Thank you for answering my questions.

---

### Official Review · Reviewer_9B3k · 2023-07-07

**Soundness:** 3 good
**Presentation:** 3 good
**Contribution:** 3 good
**Rating:** 6
**Confidence:** 4

**Summary:**

The authors propose an evaluation metric that focuses on the object-level compositionality and the explainability of the evaluation process.
They utilize off-the-shelf models including BLIPv2 (global captioner), GRiT (object detector & captioner) and GPT-4 (large language model).
In specific, they first get global caption using BLIPv2, and detect object boxes in an image along with local descriptions with GRiT, and fed the incorporation of these results with an evaluation instruction to GPT-4, which would then produce the score and the rationale according to the instruction. This way, the proposed metric produces assessments that are more correlated with the human judgements.

**Strengths:**

- The proposed metric tackles an important factor in evaluating image generative models.
- The authors reveal the potential for utilizing LLMs in evaluating image generative models.

**Weaknesses:**

- The metric would work worse for image domains that are too different from MS-COCO, which is used for training GRiT. (for example, might not work for artistic images…)
- The evaluation process consists of many sub-processes, which make it prone to have error accumulation issue.
- The metric utilizes GPT-4, which is not easily accessed by many researchers.

**Questions:**

- How many objects are selected from GRiT? Are there any rules?
- Isn't there any score variations according to the change in an evaluation instruction?
- How much the performance is degraded when the rule-enhanced rating is not used?

**Limitations:**

please refer to the Weaknesses section.

---

> ### Author Rebuttal · Authors · 2023-08-09
>
> Weaknesses
>
> 1. The experiments in our paper include a wide range of prompts and synthesized images from Stable Diffusion and DALLE models,which indicate the LLMScore’s effectiveness in evaluating text-to-image synthesis in a general domain beyond the MSCOCO domain. In addition, the local image descriptor module can be replaced with other expert models including artistic domain.
> 2. We acknowledge that the complexity of the evaluation might introduce potential error accumulation. However, LLMScore is carefully designed with a hierarchical structure that allows for both image-level and object-level visual descriptions, thereby enhancing the interpretability and granularity of the evaluation. Moreover, our substantial empirical analysis, including extensive ablation studies, demonstrates the robustness of the proposed method against such error accumulations. The high correlation of LLMScore with human judgments across various datasets (58.8% and 31.2% higher than CLIP and BLIP, respectively) further validates the effectiveness and reliability of our approach. We believe these results illustrate that the potential error accumulation has been well managed, and the innovative use of large language models within LLMScore offers a significant advancement in the text-to-image synthesis evaluation domain.
> 3. Thank you for highlighting the concern regarding the accessibility of GPT-4, which is an essential component in our LLMScore framework. We fully understand that the accessibility of GPT-4 may pose a challenge for some researchers. However, it's worth noting that LLMScore is designed to be modular and can work with other large language models that might be more readily available. The key insight of our method lies in leveraging the power of large language models for visio-linguistic compositionality evaluation, not specifically tied to GPT-4. In our revision, we provide implementation details with alternative large language models (e.g., Vicuna) and also demonstrate the core functionality. This ensures that our framework remains adaptable, extendable, and relevant to a broad research community. Therefore, we believe LLMScore is a valuable contribution to the field, regardless of specific accessibility issues related to GPT-4.
>
> Questions
>
> 1. The objects are kept if they are beyond the confidence threshold, which is calculated by multiplying the confidence of the object extractor and object descriptor. We keep the default hyperparameter setting of GRiT to extract these objects.
> 2. Yes, the LLMs will be prompted with various evaluation instructions to provide various scores.
> 3. For example, under the Concept Conjunction dataset from CompBench, w/o rule enhancement lead to 0.3660 Kendall’s tau and 0.5208 Spearmanr’s correlation with human for Stable Diffusion Model. Similar to the DALLE model, it degrades to 0.3716 and 0.5382 respectively. Though the correlation is still on par with the runner-up model. This indicates a non-trivial performance boost from our rule-enhancement. We will update this in Appendix B in the paper revision.

---

### Author Rebuttal · Authors · 2023-08-09

General Response to Ethics Reviews

Response to Ethics Reviewer kdWQ

Thanks for pointing out the issues, we address the concerns point-wisely as follow:
1. We hire amazon mechanical turk workers for evaluation. The hourly wage paid to human annotators is estimated at $15.
2. First, we will remove the prompt that is automatically detected as offensive by OpenAI DALLE API. Furthermore, we manually look at more than 30% of the samples uploaded for human evaluation to ensure no offensive images are released.
3. We will remove the logo from the figures to avoid violating the brand guidelines.

Response to Ethics Reviewer kdWQ

The architecture design is to incorporate any large language model, and that we’ve already supported other LLMs outside of OpenAI ones (e.g., Vicuna). Thus, as the reviewer suggested, we will remove the OpenAI logo and leave it as the general LLMs module.

---

### Decision · Program_Chairs · 2023-09-21

**Decision:**

Accept (poster)

**Comment:**

This paper proposes a new evaluation metric for image-text matching in text-to-image generation.
Text-image matching evalution is a very important and practical topic as the t2i has become prevalent and many t2i applications have emerged.

All reviewers appreciated for the idea, the metric quality, novelty, and writing quality, and thus gave acceptance score.

AC also agrees with the reviewers, and recommend accepting this paper.

Additionally, introducing GPT APIs requires some expense for APIs, and GPT4-API is not cheap, in particular.
AC thinks that it might be helpful for researchers to add the result on using opensource sLLM such as LLaMA2.

Please revise the paper considering the reviewers' feedback and the ethics reviewers' requests.